# Batch-Wise Permutation Feature Importance Evaluation and Problem-Specific Bigraph for Learn-to-Branch

**Yajie Niu, Chen Peng * and Bolin Liao**

College of Information Science and Engineering, Jishou University, Jishou 416000, China;
yajieniu@stu.jsu.edu.cn (Y.N.); bolinliao@jsu.edu.cn (B.L.)
* Correspondence: chen.peng@jsu.edu.cn; Tel.: +86-137-2406-2118

**Abstract:** The branch-and-bound algorithm for combinatorial optimization typically relies on a plethora of handcraft expert heuristics, and a research direction, so-called learn-to-branch, proposes to replace the expert heuristics in branch-and-bound with machine learning models. Current studies in this area typically use an imitation learning (IL) approach; however, in practice, IL often suffers from limited training samples. Thus, it has been emphasized that a small-dataset fast-training scheme for IL in learn-to-branch is worth studying, so that other methods, e.g., reinforcement learning, may be used for subsequent training. Thus, this paper focuses on the IL part of a mixed training approach, where a small-dataset fast-training scheme is considered. The contributions are as follows. First, to compute feature importance metrics so that the state-of-the-art bigraph representation can be effectively reduced for each problem type, a batch-wise permutation feature importance evaluation method is proposed, which permutes features within each batch in the forward pass. Second, based on the evaluated importance of the bigraph features, a reduced bigraph representation is proposed for each of the benchmark problems. The experimental results on four MILP benchmark problems show that our method improves branching accuracy by 8% and reduces solution time by 18% on average under the small-dataset fast-training scheme compared to the state-of-the-art bigraph-based learn-to-branch method. The source code is available online at GitHub.

**Keywords:** machine learning; combinatorial optimization; branch-and-bound; permutation feature importance

## 1. Introduction

Mixed-integer linear programming (MILP) offers a generic way to formulate and solve practical decision-making problems, e.g., routing optimization [1], manipulator control [2], and resource allocation [3]. Due to the wide applicability of MILP, numerous commercial and free MILP solvers exist, with a few well-known examples such as CPLEX [4], SCIP [5], and Gurobi [6]. The basic component of modern MILP solvers is the branch-and-bound (B&B) algorithm for global optimization [7]. Typically, B&B recursively partitions the search space by branching on the optimal solution of the linear relaxation of the MILP problem and cleverly exhausts the search space by pruning unpromising solution space until a solution with the certificate of optimality is found. The B&B algorithm relies heavily on heuristic rules, which are essentially priority guidelines devised by human experts to direct search directions toward more promising regions, such as the variable selection policy or node selection policy. Traditionally, the heuristics are carefully constructed based on expert domain knowledge and the common characteristics of specific types of problems. With its rapid development in recent years, machine learning (ML) [8] offers a way to replace some of the sophisticated hand-crafted expert heuristics in B&B [9].

To learn the variable selection policy in the B&B algorithm, Alvarez et al. adopted ML early for solving MILPs [10]. This kind of learning-based policy is also known as learn-to-branch, where learning is introduced to the optimization process to search for the

optimal solution more effectively. The branching policy is learned with many different formulated learning problems. In order to learn a partial ordering of the candidates produced by the expert, the learned policy was treated as a ranking problem in [11,12]. In [13], Alvarez et al. treated the learned policy as a regression problem and learned directly the strong branching (SB) scores of the candidates. Rather than relying on branching scores or orderings, the learned policy was treated as a classification problem and learned from expert decisions in [14]. In [15], Balcan et al. demonstrated empirically and theoretically that a high-performing branching policy could be learned for a given application domain. Learn-to-branch has become an active research area.

A key element in ML-based branch-and-bound (or ML-B&B) is state embedding, which includes the embedding of the MILP problem and its B&B solution status. In [14], a variable-constraint bipartite graph (or bigraph) representation was leveraged for B&B state embedding, and a graph convolutional neural network (GCNN) model was proposed for learning the branching policy. The bigraph representation is natural for MILPs and has shown promising performance [16,17]. However, the bigraph representation was designed for general MILP problems, i.e., aiming to apply one ML model to many different types of MILP problems. As a result, the bigraph representation contains a large number of features, which often leads to complicated ML models, as well as the extended training and inference times. Therefore, the problem-specific bigraph representation is used for each of the benchmark problems to reduce the features.

This paper aims to simplify the bigraph representation (and thus also the ML model) by problem-specific fast feature analysis and masking out non-contributing features. In ML interpretability research, a powerful tool for the feature analysis of black-box models is the permutation feature importance (PFI) measure [18]. Traditionally, the PFI is typically evaluated by permuting features over the test dataset. However, in learn-to-branch applications, the branching samples are generally collected fragmentally, large (each around 200 KB), stored as separate binary files, and are loaded and batched before being fed to the ML model. Thus, a fast feature analysis method that does not require permuting features over the whole test dataset is necessary.

Compared to the state-of-the-art methods [12–14], the contributions of this paper are summarized as follows.

1.  In order to measure the feature importance, a batch-wise PFI (BPFI) evaluation method is proposed for learn-to-branch, which permutes features within only one batch in the forward pass. The GCNN model is augmented as BPFI-GCNN by adding one shuffling switch in the GCNN model, therefore allowing the fragmented processing of the branching samples.
2.  Based on the results of the BPFI evaluation, a reduced bigraph representation is proposed for each specific benchmark problem to reduce the model complexity. The proposed representation is shown to outperform the original in most cases on both branching accuracy and solution efficiency.

The remainder of this paper is organized as follows. In Section 2, the background and related studies of ML-B&B are discussed. In Section 3, the MILPs of four NP-hard benchmark problems are introduced. In Section 4, BPFI is evaluated for the bigraph representation, according to the results of which an improvement to the bigraph representation is proposed. In Section 5, comparative experiments are carried out to verify the effectiveness of the proposed method. Finally, Section 6 concludes the paper. The source code is available online at GitHub (https://github.com/NiuYajie0/BPFI-learn2branch, accessed on 28 May 2022).

## 2. Background

### 2.1. Machine Learning Based Branch-and-Bound

Typically, B&B recursively partitions the search space by branching on the optimal solution of the linear relaxation of the MILP and cleverly exhausts the search space by pruning unpromising solution space until a solution with a certificate of optimality is found.

When branching, a candidate variable is selected as the branching variable according to the variable selection policy (or, branching policy), and two child branches are created. The branching variable is rounded down on the left child branch and rounded up on the right child branch. One of the most famous branching strategies is SB [19]. In SB, each candidate variable is tentatively branched, and the one that has the greatest product of the lower bound increase of the left branch and the lower bound increase of the right branch is selected. However, a common drawback of these expert-crafted heuristics is that they are usually time-consuming.

With the rapid development in recent years, ML offers a possibility to automatically construct effective heuristics from data by exploiting the shared structure among MILP instances [15]. In addition, using specialized deep learning and parallel computing hardware for ML models, ML-B&B can be much faster than traditional B&B implementations. Generally speaking, the training of ML models for B&B follows one of two methodologies: imitation learning (IL) and reinforcement learning (RL) [20]. In IL, the ML model is trained through the demonstration of an expert solver, such as the default MILP solver of SCIP [5]. For example, the state-of-the-art "learn to branch" method [14] frames variable selection as a classification problem and trains a GCNN using SB expert decisions as the ground truth labels. However, by the nature of IL, the IL-trained model is limited by the performance of the expert policy [21].

On the other hand, the sequential decision-making during B&B can be regarded as a Markov decision process [22], which lays the foundation for RL. By training the policy through exploration experience, RL offers a good alternative to automate the search for heuristics [23]. Therefore, it is good practice to use IL in conjunction with RL, i.e., using IL at the start of the training process, then switching to RL to continue refining the ML model. A well-known example of this practice is the AlphaGo project [24], where the experts are human players. The IL part of an IL–RL mixed training typically suffers from limited training samples; however, a good IL at the early stage can greatly improve the convergence rate of RL. Therefore, our work is focused on the performance of ML-B&B under a small-dataset fast-training scheme, which is typically the case in the early stage of an IL–RL mixed training.

### 2.2. The Bigraph Representation for State Embedding

A key element in ML-B&B is state embedding, which includes embedding the MILP problem and its B&B solution status. In previous research [12], the variable selection policy was trained offline on the collected SB scores of candidate variables. However, correlations between constraints and variables are represented by the hand-crafted features, which rely on extensive feature engineering. To address the above issue, a bigraph representation of MILP was proposed in [14], where corresponding nodes are connected if a constraint is associated with a variable, and a GCNN was used to extract useful information from the bigraph representation. This representation is natural for MILPs and has shown promising performance. In [16], Peng et al. proposed that prioritizing the sampling of certain branching decisions over others and thus providing a better branching data distribution could further improve the performance of the trained model. In [17], the authors pointed out that the GCNN-based approach relies too heavily on high-end GPU, which may not be be accessible to many practitioners. Thus, a new hybrid architecture was proposed for efficient branching on CPU machines, which combined the expressive power of GNNs with computationally inexpensive multi-layer perceptrons for branching and achieved a better balance between solution time and branching accuracy.

The original bigraph representation and its later improvements [14,17] are designed for general MILP problems, i.e., aiming to apply one ML model to as many MILP problems as possible. As a result, the bigraph representation contains a large number of features, which often leads to complicated ML models, as well as extended training and inference times. For example, in the bigraph representation [14], 13 features are used to represent a variable, 5 for a constraint, and 1 for an edge, and there can be approximately

100–1000 variables and 700–5000 constraints for a MILP instance. Therefore, this paper aims to reduce the bigraph representation by using problem-specific fast feature analysis to address the existing problem.

### 2.3. Refined Problem-Specific Branch-and-Bound

Currently, the bigraph representation in learn-to-branch is designed for general MILPs, i.e., aiming to apply one ML model to many different types of MILPs. As a result, the bigraph representation often contains a large number of features and leads to complicated ML models. Therefore, refining the bigraph representation for specific problems is an important step to further improve the ML-B&B algorithm.

In recent years, the B&B algorithm has been refined for various problems, and different methods have been proposed to utilize problem-specific knowledge. For example, in [25], a data-mining based approach was proposed to generate problem-specific knowledge for combinatorial optimization. In [26], Khachay et al. specifically designed a B&B algorithm for the precedence-constrained generalized traveling salesman problem and demonstrated that the performance of such an algorithm is competitive against the state-of-the-art MLP-solver Gurobi. Similarly, Kudriavtsev et al. proposed and refined a B&B algorithm specifically for the shortest simple path problem and demonstrated its good performance by numerical evaluations [27].

Therefore, previous studies show that the B&B algorithm has a considerable space for improvement when refined for specific problems. In this paper, the ML-B&B model is specifically refined for each of the benchmark problems using the proposed BPFI method.

## 3. Preliminaries

### 3.1. Benchmark Problems

An MILP is an optimization problem, which can be formulated as follows:

$$\arg \min_{\mathbf{x}} \left\{ \mathbf{c}^\top \mathbf{x} \mid \mathbf{A}\mathbf{x} \leq \mathbf{b}, \mathbf{l} \leq \mathbf{x} \leq \mathbf{u}, \mathbf{x} \in \mathbb{Z}^p \times \mathbb{R}^{n-p} \right\}, \tag{1}$$

where $\mathbf{c} \in \mathbb{R}^n$ denotes the objective coefficient vector, $\mathbf{A} \in \mathbb{R}^{m \times n}$ the matrix of constraint coefficients, and $\mathbf{b} \in \mathbb{R}^m$ the vector of the right-hand-sides of constraints, respectively. In addition, $\mathbf{l}, \mathbf{u} \in \mathbb{R}^n$ are the vectors of lower and upper bounds of variables, and $p \leq n$ is the number of integer variables. As popular benchmarks, four classes of MILPs are evaluated in this paper, namely set covering (SC), combinatorial auction (CA), maximum independent set (MIS), and capacitated facility location (CFL). Specifically,

1. The SC problem can be formulated as follows:

$$\min \sum_{j=1}^{n} c_j x_j,$$

$$\text{s.t.} \ \sum_{j=1}^{n} a_{ij} x_j \geq e, \quad i = 1, \ldots, m, \tag{2}$$

$$x_j \in \{0, 1\}, \quad j = 1, \ldots, n,$$

where $A = \{a_{ij}\}$ is an $m \times n$ binary matrix, and if column $j$ covers row $i$, $a_{ij} = 1$; otherwise, $a_{ij} = 0$. Define $e = (1, \ldots, 1)$, which has $m$ components, and $c_j$ is the cost of column $j$. If column $j$ is in the solution, $x_j = 1$; otherwise, $x_j = 0$.

2.  The CA problem can be formulated as follows:

$$\max \sum_{i=1}^{n} \sum_{j=1}^{m} y_{ij} w_{ij},$$

$$\text{s.t} \sum_{i=1}^{n} w_{ij} y_{ij} \leq W, \quad j = 1, \ldots, m,$$

$$\sum_{j=1}^{m} y_{ij} = 1, \quad i = 1, \ldots, n, \tag{3}$$

$$y_{ij} \in \{0, 1\}, \quad i = 1, \ldots, n \text{ and } j = 1, \ldots, m,$$

where $n, m$ are the numbers of distinct items and bidders, respectively, and $y_{ij}$ represents a binary decision variable indicating whether item $i$ is sold to bidders. The highest price that bidder $j$ with the purchasing power $W$ can offer for item $i$ is $w_{ij}$.

3.  The MIS problem can be formulated as

$$\max \sum_{v \in V} x_v,$$

$$\text{s.t. } x_u + x_v \leq 1, \quad (u, v) \in E, \tag{4}$$

$$x_v \in \{0, 1\}, \quad v \in V,$$

where $V, E$ denote the set of vertices and edges of an undirected graph, respectively, and $x_v$ for each node $v \in V$ is a binary decision variable indicating whether $v$ is selected in an independent set.

4.  The CFL problem can be formulated as

$$\min \sum_{i=1}^{n} \sum_{j=1}^{m} c_{ij} d_j x_{ij} + \sum_{i=1}^{n} f_i y_i,$$

$$\text{s.t. } \sum_{i=1}^{n} x_{ij} = 1, \quad j = 1, \ldots, m,$$

$$\sum_{j=1}^{m} d_j x_{ij} \leq u_i y_i, \quad i = 1, \ldots, n, \tag{5}$$

$$x_{ij} \geq 0, \quad i = 1, \ldots, n \text{ and } j = 1, \ldots, m,$$

$$y_i \in \{0, 1\}, \quad i = 1, \ldots, n,$$

where $c_{ij}$ is the transportation cost between customer $j$ and facility $i$, $d_j$ is the demand for customer $j$, and $x_{ij}$ is the fraction of the demand of client $j$ met from facility $i$. If facility $i$ is open, $y_i = 1$; otherwise, $y_i = 0$, and $f_i$ is the fixed cost.

*3.2. Metrics*

In this paper, two groups of metrics are used for testing the branching accuracy of an ML brancher and evaluating the solution efficiency of the solver that adopts the brancher. Specifically,

1.  The branching accuracy is described by four metrics, i.e., the percentage of times the decision has the highest strong branching score (acc@1), one of the three highest (acc@3), one of the five highest (acc@5), and one of the ten highest (acc@10) strong branching scores.

2.  The solution efficiency is described by two metrics, i.e., the 1-shifted geometric mean of the solving times in seconds (Time) and final node counts of instances (Nodes).

## 4. Methodology

In this paper, to reduce model complexity, the bigraph representation is refined according to the evaluated importance of features. In ML studies, PFI is an effective approach to gain insights into black-box models. In learn-to-branch, however, branching samples are usually large and collected fragmentally, which makes traditional PFI evaluation infeasible. To address this issue, the BPFI method is proposed to identify non-contributing features in the full bigraph representation. According to the BPFI results, a bigraph representation is designed for each of the benchmark problems.

As shown in Figure 1, BPFI evaluation is implemented by adding only one shuffling switch in the learning model. According to the BPFI evaluation, the problem-specific bigraph is built, and the non-contributing features are masked out to refine the model.

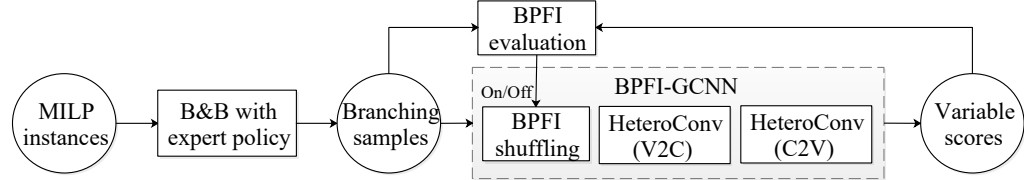

**Figure 1.** The overall BPFI framework is composed of three parts: B&B solver, BPFI evaluation, and BPFI-GCNN.

### 4.1. Batch-Wise Permutation Feature Importance

In PFI, the utility of a feature is measured by the decrease in model performance caused by permuting this feature over the dataset. The general steps for computing PFI are as follows.

1. Train and evaluate the model for a performance score $A$.
2. Evaluate the model on a modified test dataset with feature $i$ shuffled. Compute performance score $A_{i,s}$, $s = 1, \ldots, N$, for $N$ different shuffling seeds.
3. The PFI $F_i$ of feature $i$ is computed as the drop of performance after shuffling:

$$F_i = A - \frac{1}{N} \sum_{s=1}^{N} A_{i,s}. \tag{6}$$

The PFI is commonly used as an interpretation method. However, the original PFI evaluation cannot be used for learn-to-branch directly. The reason is that PFI evaluation requires shuffling features over the entire test dataset (see Figure 2a); whereas, in learn-to-branch, the branching samples are generally collected fragmentally, large (each around 200 KB), and stored as separate binary files, which makes the original PFI evaluation infeasible.

Therefore, to compute feature importance for the bigraph representation considering the fragmented branching samples dataset, the BPFI evaluation is proposed, which permutes features within only one batch in the forward pass (see Figure 2b). Formally, let $\mathcal{A}(\mathcal{M}, \mathcal{D})$ denote the performance function that computes the score of model $\mathcal{M}$ given dataset $\mathcal{D}$, and let $\mathcal{P}_i(\mathcal{D}, b)$ denote a per-batch permutation function that permutes feature $i$ of dataset $\mathcal{D}$ for a batch size $b$. Then, the performance scores $A_{i,s}$ after shuffling in the traditional PFI evaluation and $\tilde{A}_{i,s}$ in the proposed BPFI evaluation are given by

$$\begin{aligned} A_{i,s} &= \mathcal{A}(\mathcal{M}, \mathcal{P}_i(\mathcal{D}, |\mathcal{D}|)), \\ \tilde{A}_{i,s} &= \mathcal{A}(\mathcal{M}, \mathcal{P}_i(\mathcal{D}, b)), \end{aligned} \tag{7}$$

respectively, where $|\mathcal{D}|$ denotes the size of dataset $\mathcal{D}$. Since the per-batch permutation can be performed within one forward pass after a batch of samples has been loaded, BPFI evaluation is more lightweight and can approximate the traditional PFI evaluation.

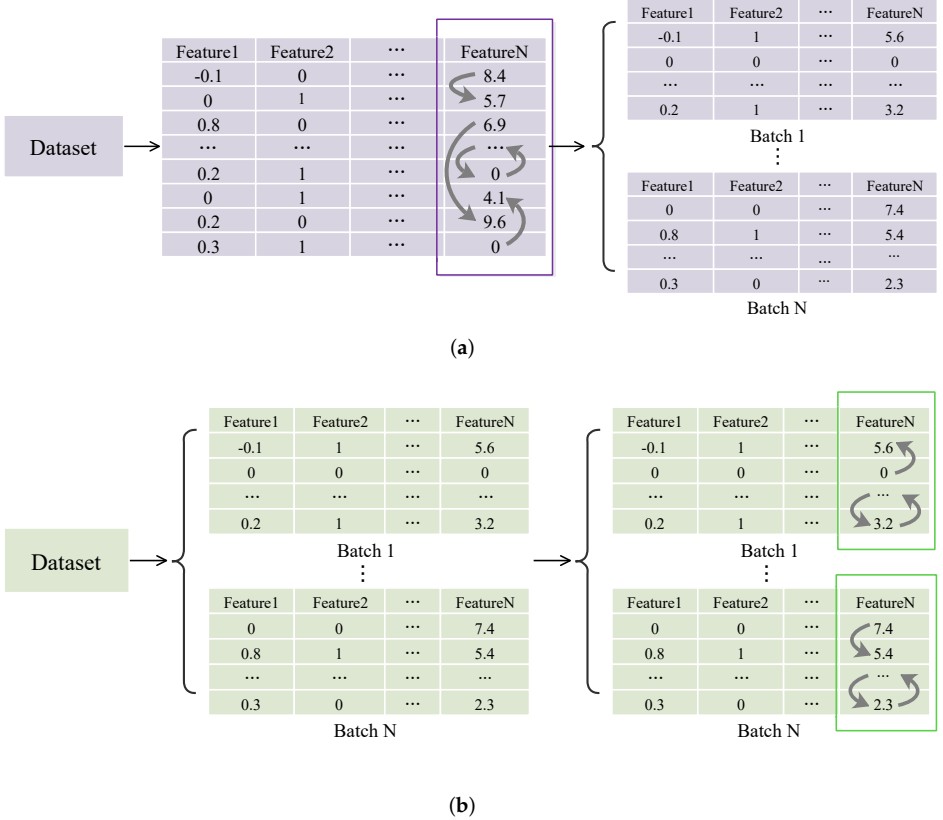

(**a**)

(**b**)

**Figure 2.** Schematic diagram of permutation feature importance evaluation and batch-wise permutation feature importance evaluation. (**a**) PFI evaluation. (**b**) BPFI evaluation.

### 4.2. Problem-Specific Bipartite Graph Representation

As shown in Figure 3, the state of the B&B process at a certain timestep can be encoded as a bigraph with node and edge features. In the bigraph, one type of node corresponds to constraints in the MILP, and the other corresponds to variables. The variable node and constraint node are connected by an edge if the variable's coefficient is non-zero in the constraint.

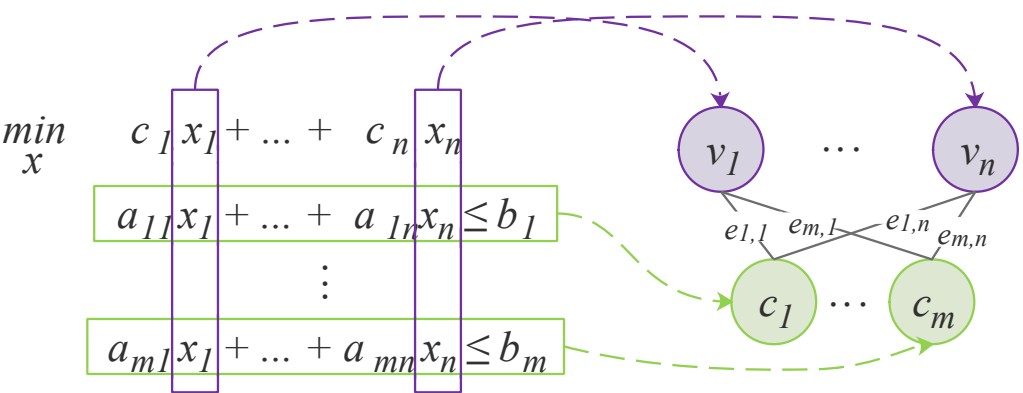

**Figure 3.** The variable-constraint bipartite graph representation of a MILP.

Given a MILP instance, let $m$ be the number of constraints of which each has $c$ features, let $n$ be the number of variables each of which has $d$ features, and each edge has $e$ features. A constraint feature matrix $\mathbf{C} \in \mathbb{R}^{m \times c}$ can be used to represent the constraint nodes, a variable feature matrix $\mathbf{V} \in \mathbb{R}^{n \times d}$ for the variable nodes, and an edge feature matrix $\mathbf{E} \in \mathbb{R}^{m \times n \times e}$ for the edges. Therefore, the original bigraph representation can be defined as $\mathbf{G} = \{\mathbf{C}, \mathbf{E}, \mathbf{V}\} \in \mathcal{G}$,

where $\mathcal{G}$ is the set of all bigraph representations of MILPs. In the proposed problem-specific bigraph representation, the non-contributing features are masked out for each of the benchmark problems. Therefore, the proposed bigraph representation can be formulated as $\mathbf{G} = \{\overline{\mathbf{C}}, \overline{\mathbf{E}}, \overline{\mathbf{V}}\}$ where $\overline{\mathbf{C}}$, $\overline{\mathbf{E}}$, and $\overline{\mathbf{V}}$ are the reduced features.

As a special heterogeneous graph, the bigraph has only two different types of nodes (constraints and variables) and two types of edges (involves-in and belongs-to). With the bipartite structure of the input graph, the graph convolution can be separated into two consecutive passes, i.e., the v-to-c and c-to-v passes, as introduced in [14]. The BPFI-GCNN further simplifies the original Full GCNN model for each problem type, according to the BPFI evaluation results. See Section 5.2 for details of the BPFI-GCNN.

## 5. Computer Experiments

In this paper, the experimental framework partially inherits from the state-of-the-art learn-to-branch project [14]. Specifically, the MILP instance generation and branching sample collection algorithms in [14] are reused, meaning that our experimental dataset is consistent with the former studies.

### 5.1. Experimental Framework

As shown in Figure 4, our experiments consist of the following six major steps.

1.  Generate instances that include the four benchmark problems, i.e., set covering, combinatorial auction, maximum independent set, and capacitated facility location.
2.  Sample the branching decision data during the B&B solution of MILP instances with SCIP 7.0 [5], obtaining branching samples datasets for training, validation, and testing.
3.  Train the GCNN model with the full bigraph representation after the shuffling switch is turned off.
4.  Perform BPFI evaluation, reduce the bigraph representation for each of the benchmark problems, and train GCNN with each reduced bigraph representation after the shuffling switch is turned on. As features are reduced, the GCNN also requires fewer parameters, thus decreasing in size.
5.  Test and compare the branching accuracy of the trained models, including the full GCNN and the BPFI-GCNN.
6.  Evaluate and compare the MILP solution efficiency of the ML-B&B models by embedding the trained GCNNs into the SCIP's B&B solution process.

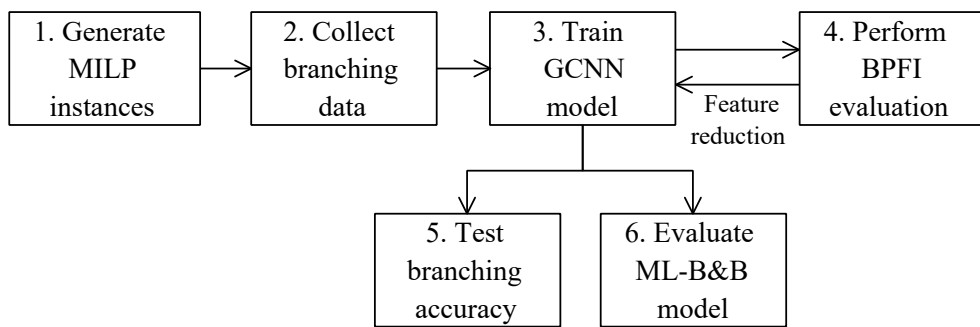

**Figure 4.** Schematic diagram of the experimental framework.

For consistency with [14], the SC instances are generated using the procedure of Balas and Ho [28] with 1000 columns for 500 (Easy), 1000 (Medium), and 2000 (Hard) rows for evaluating. The CA instances are generated using the procedure of the arbitrary relationships procedure of Leyton-Brown et al. [29] with 100 items for 500 bids (Easy), 200 items for 1000 bids (Medium), and 300 items for 1500 bids (Hard). The MIS instances are generated using the procedure of Bergman et al. [30] with 500 (Easy), 1000 (Medium), and 1500 (Hard) nodes. The CFL instances are generated using the procedure of Cornuejols et al. [31] with

100 facilities for 100 (Easy), 200 (Medium), and 400 (Hard) customers. The training and testing instances have the same size as the Easy instances.

During the BPFI evaluation, 20 independent random shufflings are performed on each feature. In the experiment, 1000 branching samples are extracted from 100 instances for training, 200 branching samples are extracted from 20 instances for validation, and the same amount is used for testing. The training process uses a batch size of 16, epoch size of 20, and max epochs of 300.

### 5.2. BPFI Evaluation and the Resulting BPFI-GCNN

The BPFI evaluation results on the four benchmark problems are shown in Figure 5, where the importance of a feature is computed as the decrease of acc@5 accuracy after this feature is shuffled. In this paper, the indicator variables are not considered in the BPFI evaluation due to their similar tensor distributions. As shown in Figure 5, it can be seen that the distribution of variable importance is different for each of the benchmark problems. Therefore, the problem-specific bigraph representation is employed based on the principle of feature reduction, i.e., a reduced bigraph representation is formalized for each of the benchmark problems. In each reduced bigraph representation, most of the non-contributing features with negative variable importance are masked out to maximize the performance of the BPFI-GCNN model.

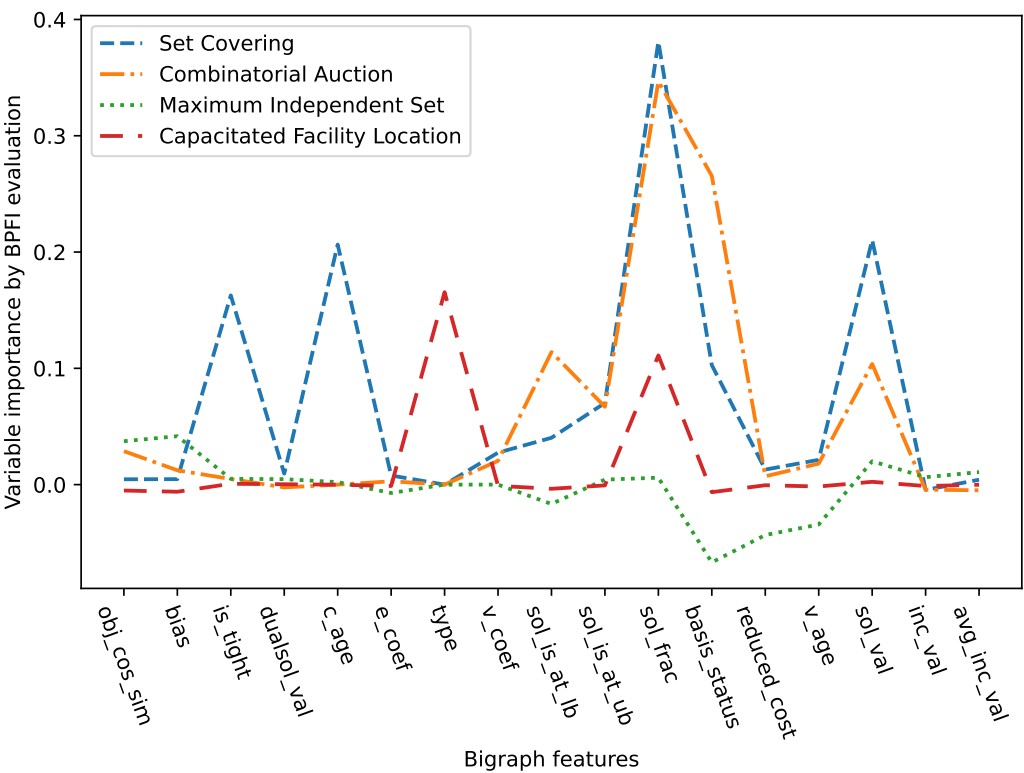

**Figure 5.** Permutation feature importance by BPFI evaluation of the bigraph features on the four MILP benchmark problems. The bigraph features are described in the supplemental file of [14] and are also detailed in Table A1 in the Appendix A for completeness.

For example, since the BPFI evaluation shows that edge features are unimportant for all four benchmark problems, the convolution in the BPFI-GCNN implementation ignores all edge weights. In addition, the BPFI-GCNN is further optimized with the Deep Graph Library (DGL) [32].

### 5.3. Comparison of Branching Accuracy

In this subsection, the branching accuracy of the full GCNN model [14] and the BPFI-GCNN model are compared. Moreover, two other ML branchers are also tested, i.e., the learning-to-score approach of Alvarez et al. [13] (TREES) based on an ExtraTrees model [33] and the learning-to-rank approach of Hansknecht et al. [12] (LMART) based on a LambdaMART model [34]. In Table 1, the branching accuracy of these models are shown under the small-dataset fast-training scheme over five seeds. It can be seen from Table 1 that the BPFI-GCNN model has the highest branching accuracy measured by these four indicators (acc@1, acc@3, acc@5, acc@10) in the four benchmark problems. Specifically, compared to the state-of-the-art bigraph-based method [14], these four branching accuracy indicators, i.e., acc@1, acc@3, acc@5 and acc@10, have increased by 8.4%, 7.5%, 7.8%, and 7.4% on average, respectively.

**Table 1.** Branching accuracy of trained ML-B&B models on testing datasets.

| Problem | Accuracy Level | Model | | | |
|---|---|---|---|---|---|
| | | **TREES** | **LMART** | **Full GCNN** | **BPFI-GCNN** |
| Set Covering | acc@1 | $46.5 \pm 1.3$ | $49.6 \pm 3.0$ | $61.7 \pm 1.4$ | $\mathbf{63.9} \pm 1.2$ |
| | acc@3 | $64.8 \pm 1.0$ | $66.5 \pm 3.7$ | $78.4 \pm 0.9$ | $\mathbf{79.3} \pm 1.4$ |
| | acc@5 | $75.8 \pm 2.3$ | $73.9 \pm 5.0$ | $87.0 \pm 1.8$ | $\mathbf{87.9} \pm 1.4$ |
| | acc@10 | $88.6 \pm 2.5$ | $84.3 \pm 6.0$ | $95.7 \pm 0.9$ | $95.2 \pm 0.7$ |
| Combinatorial Auction | acc@1 | $39.6 \pm 4.7$ | $45.8 \pm 3.0$ | $52.4 \pm 1.1$ | $\mathbf{55.9} \pm 1.8$ |
| | acc@3 | $61.1 \pm 5.6$ | $65.9 \pm 2.2$ | $75.8 \pm 1.0$ | $\mathbf{76.3} \pm 1.4$ |
| | acc@5 | $74.0 \pm 5.6$ | $85.9 \pm 0.8$ | $85.9 \pm 0.8$ | $85.6 \pm 0.9$ |
| | acc@10 | $88.7 \pm 3.8$ | $86.0 \pm 1.8$ | $94.9 \pm 0.7$ | $\mathbf{95.1} \pm 0.2$ |
| Maximum Independent Set | acc@1 | $26.1 \pm 3.5$ | $34.3 \pm 9.8$ | $29.4 \pm 26.9$ | $\mathbf{53.3} \pm 0.8$ |
| | acc@3 | $35.9 \pm 4.9$ | $47.4 \pm 8.6$ | $40.8 \pm 36.6$ | $\mathbf{68.2} \pm 0.9$ |
| | acc@5 | $40.4 \pm 5.2$ | $53.0 \pm 7.6$ | $45.8 \pm 38.9$ | $\mathbf{74.1} \pm 1.3$ |
| | acc@10 | $45.2 \pm 5.3$ | $58.7 \pm 9.8$ | $53.3 \pm 39.2$ | $\mathbf{81.3} \pm 1.9$ |
| Capacitated Facility Location | acc@1 | $55.8 \pm 2.1$ | $62.2 \pm 2.3$ | $67.2 \pm 2.2$ | $\mathbf{71.1} \pm 1.0$ |
| | acc@3 | $88.7 \pm 2.4$ | $90.7 \pm 0.9$ | $91.2 \pm 0.8$ | $\mathbf{92.6} \pm 0.2$ |
| | acc@5 | $95.7 \pm 1.9$ | $97.7 \pm 0.6$ | $96.8 \pm 0.8$ | $\mathbf{98.9} \pm 0.2$ |
| | acc@10 | $100.0 \pm 0.0$ | $100.0 \pm 0.0$ | $100.0 \pm 0.0$ | $100.0 \pm 0.0$ |

### 5.4. Comparison of Problem-Solving Efficiency

In this subsection, ML-B&B models are obtained to solve problem instances by embedding the trained models into the SCIP's B&B solution process and replacing the default SCIP brancher. Five training seeds are applied to evaluate 20 new instances for each of the problem difficulties (Easy, Medium, Hard), giving a total of 100 solving attempts per problem difficulty.

As in [16], the results in this paper are presented in the form of "mean $r \pm$ std%" to avoid the dependence of results on different experimental environments, and "$r$" is the mean of Node or Time as a reference value. For example, $0.7883r \pm 6.68\%$ means that the metric is 0.7883 times the reference value, and the per-instance standard deviation is 0.0668 averaged over all instances. In the "mean $r \pm$ std%" expression, the normalized "mean" and averaged per-instance "std" value are employed in the *t*-test statistical test.

The complete experimental results are shown in Table 2. The results show that the BPFI-GCNN model achieves significantly better results (in the sense of *t*-test significance) on most of the performance metrics. Specifically, compared to [14], the solution time has been reduced by an average of 16.8% on the Easy instances, by 22.5% on the Medium instances, and by 15% on the Hard instances. Thus, the BPFI-GCNN model achieves an overall 18% reduction on the solution time.

**Table 2.** ML-B&B solution efficiency by number of visited nodes and solution time.

| Problem | Type | Model | Node | Time | T-Stats (*p*-Value) |
|---|---|---|---|---|---|
| Set Covering | Easy | Full GCNN | $1.00r \pm 9.3\%$ | $1.00r \pm 5.3\%$ | NA |
| | | TREES | $1.45r \pm 20.1\%$ | $1.17r \pm 10.9\%$ | 26.72 (0.0) |
| | | LMART | $1.42r \pm 17.8\%$ | $\mathbf{0.80}r \pm 8.1\%$ | −26.37 (0.0) |
| | | BPFI-GCNN | $\mathbf{0.99}r \pm 6.8\%$ | $0.98r \pm 4.3\%$ | −0.36 (0.7) |
| | Medium | Full GCNN | $1.00r \pm 16.6\%$ | $1.00r \pm 15.7\%$ | NA |
| | | TREES | $1.35r \pm 14.8\%$ | $1.53r \pm 12.6\%$ | 22.50 (0.0) |
| | | LMART | $1.71r \pm 27.6\%$ | $0.99r \pm 22.7\%$ | 0.52 (0.6) |
| | | BPFI-GCNN | $\mathbf{0.89}r \pm 9.0\%$ | $\mathbf{0.95}r \pm 7.9\%$ | −4.98 (0.0) |
| | Hard | Full GCNN | $1.00r \pm 31.6\%$ | $1.00r \pm 29.4\%$ | NA |
| | | TREES | $0.87r \pm 13.7\%$ | $1.17r \pm 9.3\%$ | 1.93 (0.1) |
| | | LMART | $1.42r \pm 20.3\%$ | $1.04r \pm 17.7\%$ | 0.10 (0.9) |
| | | BPFI-GCNN | $\mathbf{0.75}r \pm 8.2\%$ | $\mathbf{0.79}r \pm 6.7\%$ | −4.04 (0.0) |
| Combinatorial Auction | Easy | Full GCNN | $1.00r \pm 12.3\%$ | $1.00r \pm 8.2\%$ | NA |
| | | TREES | $1.28r \pm 29.7\%$ | $1.11r \pm 16.7\%$ | 7.03 (0.0) |
| | | LMART | $1.33r \pm 24.6\%$ | $\mathbf{0.71}r \pm 9.0\%$ | −30.69 (0.0) |
| | | BPFI-GCNN | $\mathbf{0.99}r \pm 13.2\%$ | $1.02r \pm 8.6\%$ | 2.19 (0.0) |
| | Medium | Full GCNN | $1.00r \pm 14.2\%$ | $1.00r \pm 11.5\%$ | NA |
| | | TREES | $4.0r \pm 117.9\%$ | $4.01r \pm 111.7\%$ | 8.29 (0.0) |
| | | LMART | $2.30r \pm 34.6\%$ | $1.16r \pm 28.7\%$ | 7.99 (0.0) |
| | | BPFI-GCNN | $1.01r \pm 12.0\%$ | $\mathbf{0.99}r \pm 8.9\%$ | −1.64 (0.1) |
| | Hard | Full GCNN | $1.00r \pm 19.1\%$ | $1.00r \pm 16.6\%$ | NA |
| | | TREES | $8.20r \pm 58.8\%$ | $11.01r \pm 58.8\%$ | 15.22 (0.0) |
| | | LMART | $4.11r \pm 73.2\%$ | $3.01r \pm 71.9\%$ | 10.38 (0.0) |
| | | BPFI-GCNN | $\mathbf{0.97}r \pm 14.3\%$ | $\mathbf{0.93}r \pm 13.2\%$ | −5.33 (0.0) |
| Maximum Independent Set | Easy | Full GCNN | $1.00r \pm 160.4\%$ | $1.00r \pm 102.9\%$ | NA |
| | | TREES | $0.55r \pm 71.1\%$ | $0.73r \pm 36.8\%$ | −7.46 (0.0) |
| | | LMART | $0.45r \pm 108.9\%$ | $0.41r \pm 36.5\%$ | −9.51 (0.0) |
| | | BPFI-GCNN | $\mathbf{0.20}r \pm 53.7\%$ | $\mathbf{0.36}r \pm 16.1\%$ | −9.62 (0.0) |
| | Medium | Full GCNN | $1.00r \pm 83.2\%$ | $1.00r \pm 82.1\%$ | NA |
| | | TREES | $1.93r \pm 17.7\%$ | $3.20r \pm 11.6\%$ | 5.55 (0.0) |
| | | LMART | $0.61r \pm 85.7\%$ | $0.51r \pm 77.2\%$ | −4.36 (0.0) |
| | | BPFI-GCNN | $\mathbf{0.13}r \pm 94.9\%$ | $\mathbf{0.20}r \pm 81.4\%$ | −7.75 (0.0) |
| | Hard | Full GCNN | $1.00r \pm 45.2\%$ | $1.00r \pm 24.2\%$ | NA |
| | | TREES | $0.76r \pm 9.3\%$ | $1.29r \pm 2.4\%$ | 4.26 (0.0) |
| | | LMART | $1.03r \pm 29.7\%$ | $1.06r \pm 9.0\%$ | 1.55 (0.1) |
| | | BPFI-GCNN | $0.76r \pm 36.4\%$ | $\mathbf{0.79}r \pm 30.2\%$ | −2.49 (0.01) |
| Capacitated Facility Location | Easy | Full GCNN | $1.00r \pm 23.5\%$ | $1.00r \pm 16.3\%$ | NA |
| | | TREES | $1.15r \pm 24.3\%$ | $1.50r \pm 17.5\%$ | 24.42 (0.0) |
| | | LMART | $1.10r \pm 22.7\%$ | $\mathbf{0.88}r \pm 16.6\%$ | −4.37 (0.0) |
| | | BPFI-GCNN | $\mathbf{0.99}r \pm 21.7\%$ | $0.97r \pm 15.1\%$ | −1.70 (0.09) |
| | Medium | Full GCNN | $1.00r \pm 15.0\%$ | $1.00r \pm 13.8\%$ | NA |
| | | TREES | $1.18r \pm 18.4\%$ | $1.47r \pm 16.6\%$ | 19.6 (0.0) |
| | | LMART | $1.00r \pm 17.6\%$ | $\mathbf{0.89}r \pm 14.9\%$ | −7.25 (0.0) |
| | | BPFI-GCNN | $\mathbf{0.99}r \pm 14.3\%$ | $0.96r \pm 12.8\%$ | −1.46 (0.15) |
| | Hard | Full GCNN | $1.00r \pm 16.0\%$ | $1.00r \pm 14.6\%$ | NA |
| | | TREES | $1.07r \pm 16.3\%$ | $1.28r \pm 16.1\%$ | 12.49 (0.0) |
| | | LMART | $\mathbf{0.87}r \pm 13.9\%$ | $\mathbf{0.86}r \pm 13.3\%$ | −8.64 (0.0) |
| | | BPFI-GCNN | $0.90r \pm 11.7\%$ | $0.89r \pm 12.0\%$ | −6.85 (0.0) |

## 6. Conclusions

In this paper, the BPFI evaluation method has been proposed, which allows the fragmented processing of branching samples. Based on the results of the BPFI evaluation, a refined bigraph representation for each of the benchmark problems has been proposed for the BPFI-GCNN model. The experimental results have shown that the proposed BPFI-GCNN model improves the accuracy of the B&B solution, shortening the solution time on four MILP benchmark problems.

Our work is limited to ML-B&B under a small-dataset fast-training scheme, which corresponds to the IL part of IL–RL mixed training. However, the effectiveness of the full IL–RL mixed training using this approach for IL remains to be studied. Furthermore, the explainability of GCNN for learn-to-branch is an interesting research direction that is worth exploring.

**Author Contributions:** Conceptualization, Y.N. and C.P.; methodology, C.P. and Y.N.; software, Y.N.; validation, Y.N. and C.P.; formal analysis, Y.N. and C.P.; investigation, B.L.; data curation, B.L.; writing—original draft preparation, Y.N.; writing—review and editing, Y.N. and C.P.; visualization, Y.N.; supervision, C.P.; project administration, Y.N.; funding acquisition, C.P. All authors have read and agreed to the published version of the manuscript.

**Funding:** This work is supported by the Natural Science Foundation of China under Grant 62006095 and 62066015, by the Natural Science Foundation of Hunan Province, China, under Grant 2021JJ40441, by the Research Foundation of Education Bureau of Hunan Province, China, under Grant 20B470, and by the Jishou University Graduate Research and Innovation Project XXJD202204.

**Institutional Review Board Statement:** Not applicable.

**Informed Consent Statement:** Not applicable.

**Data Availability Statement:** Not applicable.

**Conflicts of Interest:** The authors declare no conflict of interest.

## Abbreviations

The following abbreviations are used in this manuscript:

| | |
|---|---|
| MILP | Mixed-integer linear programming |
| ML | Machine learning |
| B&B | Branch-and-bound |
| GCNN | Graph convolutional neural network |
| BPFI | Batch-wise permutation feature importance |
| PFI | Permutation feature importance |
| SC | Set covering |
| CA | Combinatorial auction |
| MIS | Maximum independent set |
| CFL | Capacitated facility location |
| SB | Strong branching |
| IL | Imitation learning |
| RL | Reinforcement learning |
| DGL | Deep graph library |

## Appendix A

The features of the full bigraph representation are given in Table A1.

**Table A1.** Feature matrix **C** for the constraints, feature matrix **E** for the edges and feature matrix **V** for the variables in the bigraph representation **G** = {**C**, **E**, **V**} [14].

| Tensor | Feature | Description |
|--------|---------|-------------|
| **C** | obj_cos_sim | Cosine similarity with objective. |
| | bias | Bias value, normalized with constraint coefficients. |
| | is_tight | Tightness indicator in LP solution. |
| | dualsol_val | Dual solution value, normalized. |
| | c_age | LP age, normalized with total number of LPs. |
| **E** | e_coef | Constraint coefficient, normalized per constraint. |
| **V** | type | Type (binary, integer, impl.integer, continuous) as one-hot encoding. |
| | v_coef | Objective coefficient, normalized. |
| | has_lb | Lower bound indicator. |
| | has_ub | Upper bound indicator. |
| | sol_is_at_lb | Solution value equals lower bound. |
| | sol_is_at_ub | Solution value equals upper bound. |
| | sol_frac | Solution value fractionality. |
| | basis_status | Simplex basis status (lower, basic, upper, zero) as one-hot encoding. |
| | reduced_cost | Reduced cost, normalized. |
| | v_age | LP age, normalized. |
| | sol_val | Solution value. |
| | inc_val | Value in incumbent. |
| | avg_inc_val | Average value in incumbents. |

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
