# Peer review of "Batch-Wise Permutation Feature Importance Evaluation and Problem-Specific Bigraph for Learn-to-Branch"

_electronics, doi:10.3390/electronics11142253_

Round 1

Reviewer 1 Report

Please see the attached pdf.

Reviewer 2 Report

Learn-to-branch uses bigraph representation for state encoding. The paper selects the most important features to reduce the model size by proposing using batch-wise permutation feature. Problem-specific  bigraph features are then learned. It is illustated empirically that BPFI-GCNN(current work) performs better than Full GCNN.

Here are some concerns that I have about the manuscript that hopefully would improve the work.

1.  It would be great to make the paper self-contained. You are encouraged to include an example of how to convert MILP to a bigraph.

2. How do we decide the batches? If we decide the batches differently, would the solution quality changes?

3. By learning at problem-level, the quality of the solution should improve. However, how do we ensure that the instances used are representative of the problem?

4. Explain the source of the data. Are the data generated randomly or are they from a data reposisory? How do the quality of the solutions change if the data doesn't come from the same distribution?

5. How do you determines the batch size? How does batch size affects your result?

Reviewer 3 Report

In this paper, the BPFI evaluation method has been proposed which allows fragmented processing of the branching samples. After going through the paper, I found some concerns, as listed below:

1.      A concise and factual abstract is required to state briefly the purpose of the research, the principal results and the major conclusions. Add some of the most important quantitative results to the abstract. Focus on the advantages of the proposed method with respect to the obtained results.

2.      Most of the ideas written were already described in many literatures. The Authors tried to compile it but lack of the enhancement of the interrelation analysis between the references. It is advised that the authors give a deeper analysis on how these ideas become more applicative strategies so that they can contribute to the next step of implementation.

3.      The novelty of the present work should be well stated and justified. The new author's contribution should be justified regarding the previous works in the literature.

4.      Avoid lumping references such as [8-10] and [11-14]. Instead summarize the main contribution of each referenced paper in a separate sentence and by including the reference number.

5.      I would like to see more discussion of the literature so that I can clearly identify the article relates to competing ideas.

6.      The problem formulation should be modified to clearly demonstrate the model, and if possible, a simple example should be added to demonstrate the idea of this paper.

7.      Future scope, current limitation must be discussed, for example a short paragraph may be included in the conclusion section more explicitly.

8.      There are several grammatical mistakes. Please work close to a native English speaker to refine the language of this paper.

Round 2

Reviewer 1 Report

Thank you for addressing most of my comments. Please see the attached pdf for more details.

Perhaps the most important issues are as follows:

1. You copied a table from another paper into their own (see the attached pdf for more details). The table only contains notation and it is not actually used in the paper, but as written it does seem to be plagiarized. 

2. Also, you authors still do not provide any code or another method for someone to reproduce your results. As such, it does not seem possible to verify your results. 

Author Response

Thank you for your constructive comments. Please see the attached pdf.

Reviewer 3 Report

 Accept in present form

Author Response

Thank you for your time, efforts, and recognition given to our manuscript.

Round 3

Reviewer 1 Report

Thank you for addressing my comments. There is only one typo (that I introduced in my last  review... sorry). Line 160 should say 'the B&B algorithm has considerable space for improvement'. 

I think the paper is in a good spot to be accepted.